# Epigenetic Modifiers as Novel Therapeutic Targets and a Systematic Review of Clinical Studies Investigating Epigenetic Inhibitors in Head and Neck Cancer

**DOI:** 10.3390/cancers13205241

**Published:** 2021-10-19

**Authors:** Kyunghee Burkitt, Vassiliki Saloura

**Affiliations:** 1Head and Neck Medical Oncology, University Hospitals Cleveland Medical Center, Case Western Reserve University School of Medicine, Cleveland, OH 44106, USA; 2Thoracic and GI Malignancies Branch, Center for Cancer Research, National Cancer Institute, Bethesda, MD 20892, USA

**Keywords:** epigenetic modifiers, head and neck squamous cell carcinoma, histone acetylation/deacetylation, histone methylation/demethylation, DNA methylation/demethylation

## Abstract

**Simple Summary:**

Head and neck cancer is the sixth most common malignancy worldwide and it affects approximately 50,000 patients annually in the United States. Current treatments are suboptimal and induce significant long-term toxicities that permanently affect quality of life. Novel therapeutic approaches are thus urgently needed to increase the survival and quality of life of these patients. Epigenetic modifications have been recognized as potential therapeutic targets in various cancer types, including head and neck cancer. The objective of this review is to provide a brief overview of the function of important epigenetic modifiers in head and neck cancer, and to discuss the results of past and ongoing clinical trials evaluating epigenetic interventions targeting these epigenetic modifiers in head and neck cancer patients. The field of epigenetic therapy in head and neck cancer is still nascent; however, it holds significant promise. Although more specific epigenetic drugs are being developed, we envision the rational design of clinical trials that will target a select group of head and neck cancer patients with epigenetic vulnerabilities that can be targeted in combination with immunotherapy, chemotherapy and/or radiotherapy, rendering higher and durable responses while minimizing chronic complications for patients with head and neck cancer.

**Abstract:**

The survival rate of head and neck squamous cell carcinoma patients with the current standard of care therapy is suboptimal and is associated with long-term side effects. Novel therapeutics that will improve survival rates while minimizing treatment-related side effects are the focus of active investigation. Epigenetic modifications have been recognized as potential therapeutic targets in various cancer types, including head and neck cancer. This review summarizes the current knowledge on the function of important epigenetic modifiers in head and neck cancer, their clinical implications and discusses results of clinical trials evaluating epigenetic interventions in past and ongoing clinical trials as monotherapy or combination therapy with either chemotherapy, radiotherapy or immunotherapy. Understanding the function of epigenetic modifiers in both preclinical and clinical settings will provide insight into a more rational design of clinical trials using epigenetic interventions and the patient subgroups that may benefit from such interventions.

## 1. Introduction

Epigenetic modifications have attracted significant interest in cancer research. In contrast to genetic alterations, epigenetic modifications influence gene expression without permanent changes in the genomic sequences. Epigenetic modifications are known to play an important role in the progression of cancer cells and also in the interactions between tumor cells and the tumor microenvironment [1]. Given the importance of epigenetic modifications in cancer progression, epigenetic interventions have been recognized as potential therapeutic strategies for the treatment of cancer either as monotherapy or as combination treatment with other drugs.

Head and neck squamous cell carcinoma (HNSCC) is the sixth most common cancer worldwide [2]. Particularly in human papilloma virus (HPV)-negative HNSCC, despite aggressive treatment in the curative-intent setting, the 5-year overall survival (OS) remains low (39.7%), and this has not been improved significantly in the past 30 years. Platinum-based combination chemotherapy has been used for many decades as part of the standard of care therapy for both locoregionally advanced and recurrent/metastatic (R/M) HNSCC.

However, the majority of patients become resistant to chemotherapy and their disease progresses. Recently, the FDA approved pembrolizumab, an anti-programmed-death ligand 1 (PD-1) antibody, as a first-line immunotherapy treatment for R/M HNSCC; however, the majority of patients do not respond [3]. For this reason, efforts to investigate novel drugs that will improve both chemotherapy and immunotherapy responses in HNSCC are ongoing. Within this context, preclinical and clinical efforts investigating epigenetic drugs either as monotherapy or in combination with chemotherapy or immune therapy have been evolving.

In this review, we summarize preclinical data on the dysregulation of main categories of epigenetic regulators, and review previous and ongoing clinical trials evaluating the role of available epigenetic drugs in HNSCC as monotherapy and in combination with chemotherapy, radiotherapy or immunotherapy.

## 2. Materials and Methods

The PRISMA guidelines for systematic reviews were followed. We used the PubMed literature database and clinicaltrials.gov to systematically investigate the literature and identify original research articles and clinical trials that investigate the role of epigenetic regulators in HNSCC. The following search terms were used: “epigenetic modification”, “HDAC inhibitor”, “DNMT inhibitor”, “Bromodomain and extra terminal domain (BET) proteins”, “immunotherapy”, “chemotherapy” and/or “radiation therapy”, “head and neck cancer”. We focused on studies published in peer-reviewed journals and excluded studies that focused exclusively on nasopharyngeal carcinoma or salivary gland cancers, given the different biology of these cancer types compared to HNSCC. We categorized the studies based on the epigenetic mechanisms targeted and subdivided each section into a “preclinical rationale” section and a “clinical trials” section, which describes the completed and ongoing clinical trials/concepts that involve epigenetic inhibitors in HNSCC. Given the distinct biology between HPV-positive and HPV-negative HNSCC, we specified whether each study included HPV-positive and/or HPV-negative patients or both. With the exception of two studies (Table 1), all other studies were agnostic to HPV-status.

## 3. DNA Methylation

### 3.1. DNA Methylation and Preclinical Rationale for Using DNMT Inhibition in HNSCC

DNA methylation pertains to the methylation of cytosine bases in the DNA, which is catalyzed by DNA methyltransferases (DNMTs). It regulates the balance between open and closed chromatin, and the rate of DNA methylation is inversely proportional to transcription [4]. Aberrations in DNA methylation and epigenetic gene silencing affect cell proliferation, apoptosis, differentiation, cell cycle and tumorigenesis [5]. There are three DNMTs: DNMT1, 3A and 3B. Overexpression of these DNMTs in different types of tumors results in hypermethylation and oncogenic activation [6]. DNMT overexpression is associated with aberrant DNA methylation in solid tumors, resulting in lymph node metastasis and poor prognosis in cancer patients [7,8,9].

In HNSCC, specifically in laryngeal squamous cell carcinoma, the frequent hypermethylation of genes that are involved in cellular proliferation, apoptosis (*DAPK, RASSF1A, RARbeta*) and DNA repair (MGMT) was observed, and the hypermethylation of O6-methylguanine-DNA methyltransferase (MGMT) was associated with lymph node metastasis [10]. In addition, the mRNA expression of DNA methyltransferases DNMT1, DNMT3A and DNMT3B was upregulated in 36.9%, 26% and 23% of 65 oropharyngeal squamous cell carcinoma patients (OSCC, HPV status was not specified in this study), respectively [11]. DNMT1 overexpression was negatively correlated with the overall survival and relapse-free survival of patients. More specifically, patients with DNMT1 overexpression had a ~2.4-fold higher risk to relapse than those with lower expression. The study suggested that DNMT1 gene expression could be a potential prognostic marker and epigenetic target for the treatment of OSCC.

Chen et al. showed that DNMT3B is involved in the induction of the epithelial-to-mesenchymal transition (EMT) phenotype in HPV-negative HNSCC cell lines [12]. DNMT3B was upregulated in invasive HNSCC cell lines, methylating the promoter of *E-cadherin* and inhibiting its expression. Knockdown of DNMT3B by siRNA interference was shown to reduce EMT and cell invasion. Further in vivo experimental validation is warranted to support DNMT3B as a potential therapeutic target to inhibit invasion and metastasis in HNSCC. Decitabine, a DNMT inhibitor, has been shown to reverse methylation and restore cisplatin sensitivity in in vitro and in vivo models of cisplatin-resistant HPV-negative HNSCC [13]. The study specifically looked at six genes (*CRIP1*, *G0S2*, *MLH1*, *OP3*, *S100*, and *TUBB2A*) which are known to be hypermethylated in cisplatin-resistant cancer cell lines. The authors showed that decitabine treatment of cisplatin-resistant HNSCC cells resulted in promoter demethylation and increase in gene expression of *CRIP1*, *G0S2*, *MLH1* and *TUBB2A*, restoring cisplatin sensitivity. Combination treatment of cisplatin and decitabine significantly reduced tumor growth in a cisplatin-resistant tongue squamous cell carcinoma xenograft.

Additionally, De Schutter et al. [10] showed that decitabine with or without HDAC inhibition radiosensitized four out of six HPV-negative HNSCC cell lines [14], inducing increased apoptosis, radiation-induced G_2_/M phase arrest and γH2AX formation. Although the authors attempted to analyze the promoter methylation and acetylation status of a panel of 15 genes involved in DNA repair and cell cycle regulation processes, no mechanisms were determined that could explain the radiosensitizing effects of DNMT and HDAC inhibition. The role of DNA demethylation through azacytidine has also been investigated in HPV-positive HNSCC [15]. Azacytidine induced growth inhibition and cell death, reduced the expression of HPV genes, stabilized p53 and induced p53 dependent apoptosis in HPV-positive HNSCC cells. Furthermore, azacytidine suppressed the expression and activity of matrix metalloproteinases (MMPs) in HPV-positive HNSCC, and also inhibited tumor growth and invasion in HPV-positive xenograft tumors.

The above preclinical studies suggest a potential clinical therapeutic benefit of using DNMT inhibitors in HNSCC, as discussed below.

### 3.2. Clinical Trials with DNMT Inhibitors as Monotherapy or in Combination with Chemotherapy or Immunotherapy in HNSCC

Currently, azacytidine and decitabine are FDA-approved DNMT inhibitors for the treatment of myelodysplastic syndrome and acute myeloid leukemia [16,17]. In this section, we review ongoing clinical trials using DNMT inhibitors as monotherapy and also in combination with either chemotherapy or immunotherapy in HNSCC. (Table 1). These clinical trials are ongoing; therefore, results are currently pending.

#### 3.2.1. Azacytidine

Based on preclinical data described above [15], a window of opportunity, phase 2 clinical trial (NCT02178072, T-tare) was initiated and is still open at the Yale Cancer Center to assess the biological effects and safety of singe-agent azacytidine administered intravenously at 75 mg/m^2^/d for 5 or 7 days in HPV-positive HNSCC patients. Initially, the trial also included HPV-negative patients, although it was later amended to include only HPV-positive patients due to ensuing evidence of the more potent biological activity of azacytidine in this subgroup of HNSCC. Patients with newly diagnosed, surgically resectable HNSCC are eligible. The primary objective of this study is to determine the proportion of HPV-positive patients in whom azacytidine increases APOBEC RNA expression. Secondary objectives are: (1) to investigate the proliferation, apoptosis and reactivation of IFN pathways in patients with azacytidine; (2) to investigate the clinical activity of azacytidine; and (3) to investigate the safety of azacytidine. Preliminary results from the analysis of five HPV-positive tumors from patients participating in this window of opportunity study showed that after 5 or 7 days of treatment, azacytidine decreased the expression of HPV genes by approximately 2–5-fold, stabilized and increased the expression of p53, and induced the activation of caspase 3 and apoptosis in HPV-positive HNSCC tumors. Similar results were observed in HPV-positive HNSCC cell lines. Furthermore, treatment with azacytidine activated type I IFN responses in some HPV-positive HNSCC cell lines, repressed the expression of matrix metalloproteinases (MMPs) and deterred the blood vessel invasive ability of HPV-positive HNSCC xenograft tumors. These data suggest that demethylation therapy could be an effective therapeutic intervention in HPV-positive HNSCC.

#### 3.2.2. Decitabine

Intravenous decitabine is being evaluated as monotherapy in the treatment of HPV-positive anogenital and HNSCC patients after radiotherapy or as late salvage (NCT04252248, DERANO trial). This is a phase 1 study to evaluate the safety and tolerability, and the first signs of efficacy of a decitabine regimen in two strata of patients with HPV-positive anogenital and HNSCC. Stratum 1 consists of patients at high risk of disease recurrence, and stratum 2 consists of patients that have failed or refused standard therapy in the R/M setting. Patients are treated with intravenous decitabine infusion at 20 mg/m^2^ daily for 5 days, starting on day 1 with a single repetition of a cycle on day 29. The duration of the trial for each patient is expected to be 6 months (two 28-day cycles of decitabine plus four months of additional follow up). Primary endpoint is the incidence of dose-limiting toxicities. Secondary endpoints are the objective response rate (ORR), disease control rate (DCR), quality of life, OS (assessed ≥6 months) and progression-free survival (PFS). Results from this study are still pending.

Oral decitabine (ASTX727) is currently also being evaluated in combination with durvalumab in R/M HNSCC patients (NCT03019003). This is a non-randomized, open-label, phase Ib/2 study to assess the safety and efficacy of oral decitabine (ASTX727) and durvalumab (MEDI4736) in combination. Inclusion criteria include R/M HNSCC (oral cavity, oropharynx, hypopharynx, or larynx) that have progressed during or after treatment with anti-PD-1, anti-PD-L1 or anti-CTLA4 monotherapy. Oral decitabine is administered alone in cycle 1 and the combination of oral decitabine and durvalumab is given in cycles 2–12. The primary objective for the phase Ib part of the study is to determine the biologically effective dose of oral decitabine, as defined by changes in HLA class I and tumor antigen expression, whereas the secondary endpoint is the incidence of adverse treatment-related events. The primary objective for the phase 2 part of the study is to determine the 2-year PFS, whereas secondary endpoints include the best overall ORR and 2-year OS. Results from this study are also pending.

## 4. Histone Modifications

Histone modifications play an important role in modifying the chromatin structure and DNA transcriptional activity. Dysregulation in histone modifications is known to be associated with the initiation and progression of cancer [18]. There are a plethora of different histone modifications, but the most studied are histone acetylation/deacetylation and histone methylation/demethylation. Table 2 summarizes examples of histone deacetylase inhibitors, including their classification, specificity and those that are FDA-approved for cancer treatment. Here, we briefly review preclinical studies investigating the role of histone acetylation/deacetylation and methylation/demethylation in the tumorigenesis of HNSCC, and provide an overview of clinical trials using currently available drugs targeting these histone modifications (Table 3).

### 4.1. Histone Acetylation/Deacetylation and Preclinical Rationale for Using HDAC Inhibition in HNSCC

The acetylation and deacetylation of histones can induce conformational changes of nucleosomes and are catalyzed by histone acetyltransferase (HATs) and histone deacetylases (HDACs). Acetylation results in the relaxation of chromatin, which, in turn, induces gene transcription, whereas deacetylation compacts chromatin, which results in the decreased accessibility of transcription factors to chromatin. HDACs are a class of zinc-dependent metalloenzymes and play an important role in cancer by deacetylating histone and nonhistone substrates, which are involved in various biological processes including cell cycle regulation, apoptosis, DNA-damage response, metastasis and angiogenesis [26].

Altered expression and/or function of HDACs has been observed in different types of cancer; therefore, targeting HDACs has been investigated in cancer therapy. A number of HDAC inhibitors have become available (Table 2) and are currently being investigated in clinical trials.

In a preclinical study using HPV-negative HNSCC cell lines, lower levels of global H3K9 acetylation were observed compared to normal oral keratinocytes. The pharmacological inhibition of HDACs decreased HNSCC proliferation and reduced the cancer stem cell (CSC) population [27]. This study suggests that HDAC inhibition may affect tumor “plasticity” and thereby the development of resistance to therapy. In another study, low levels of H3K9 acetylation were also shown to be positively correlated with poor survival in oral cancer [28].

A study investigating the mechanism underlying the chemoresistance of HPV-negative HNSCC cells found that activated NF-κB signaling induces chemotherapy resistance by promoting histone deacetylation. Investigators used HDAC inhibitors, which prevented NF-κB-induced cisplatin resistance and increased cytotoxicity following cisplatin treatment [29]. Another study using a pan-HDAC inhibitor, sodium phenylbutyrate, showed that it sensitizes the response of HPV-negative HNSCC cells to cisplatin and that this was mediated through disruption of the Fanconi anemia (FA)/breast cancer susceptibility protein (BRCA) pathway. Specifically, sodium phenylbutyrate treatment reduced the expression of BRCA1, and this was associated with the reduced formation of Fanconi anemia complementation group D2 (FANCD2) nuclear foci, which is a functional readout of DNA repair through the FA/BRCA pathway. Re-expression of BRCA1 restored the ability of HPV-negative HNSCC cells to form FANCD2 foci following cisplatin treatment and enhanced cisplatin resistance. Accordingly, sodium phenylbutyrate sensitized cancer cells defective in the FA pathway to cisplatin [30]. Consistently, another study showed that the depletion of HDAC1 and 2 in cisplatin-resistant cells reversed cisplatin resistance and decreased tumorsphere formation [31]. HDACs were overexpressed in HPV-negative HNSCC tumors as well as cisplatin-resistant HPV-negative HNSCC cell lines. In addition, using an SCID mouse xenograft model of HNSCC, suberoylanilide hydroxamic acid (SAHA), an HDAC inhibitor, significantly enhanced the anti-tumor activity of cisplatin treatment with no additional systemic toxicity and significantly decreased tumor metastasis and *NANOG* expression, a marker of stemness. Finally, He et al. recently showed that HDAC inhibition may also suppress the proliferation, migration and invasion of HPV-negative HNSCC cells through the selective action of HDAC inhibitors on the EGFR-ADP ribosylation factor (Arf1) complex axis [32]. Interestingly, the authors found that treatment of HNSCC cells with HDAC inhibitors significantly reduced global tyrosine phosphorylation levels, and particularly decreased the phosphorylation levels of EGFR by half. HDAC inhibition also decreased the total EGFR protein amounts and the activation of Arf1, which requires its interaction with phosphorylated EGFR. The authors concluded that HDAC inhibition suppresses the invasive and migratory potential of HPV-negative HNSCC through disruption of the EGFR-Arf1 complex pathway.

### 4.2. Clinical Trials with HDAC Inhibitors in HNSCC

The above preclinical data suggest that HDAC inhibition may sensitize HPV-negative HNSCC cells to cisplatin, and may suppress the proliferative capacity, and the migratory and invasive potential of HPV-negative HNSCC cells.

Different HDAC inhibitors have been evaluated in clinical trials as monotherapy, in combination with chemoradiation, and more recently, immunotherapy in HNSCC. In this section, we review previously completed and ongoing clinical trials using HDAC inhibitors as monotherapy and combination therapy with either chemotherapy or immunotherapy in HNSCC. Table 3 summarizes the results of these clinical trials.

#### 4.2.1. HDAC Inhibitors as Monotherapy

Romidepsin, a cyclic peptide, was evaluated in a phase 2 clinical trial in 14 patients with R/M HNSCC that had received any number of lines of systemic chemotherapy (NCT00084682) [19]. The purpose of the study was to evaluate the efficacy and the in vivo pharmacodynamic effects in tumors and normal adjacent tissues in HNSCC patients. Patients agreed to provide pre- and post-treatment samples of accessible tumor and oral mucosa, as well as blood samples. The HPV status of evaluated patients in the study was not specified. Patients received romidepsin at 13 mg/m^2^ intravenously over 4 h on days 1, 8, and 15 of 28-day cycles, with response assessments by RECIST every 8 weeks. Thirteen patients were evaluated for their response to romidepsin. Objective responses were not observed; however two heavily pretreated patients exhibited brief clinical disease stabilization. Pharmacodynamic effects of HDAC inhibition, such as histone H3 hyperacetylation in peripheral blood mononuclear cells (PBMCs), and the induction of p21 and decreased Ki67 staining in tumor samples, were observed in seven pre- and post-treatment sample pairs, which confirmed the biological effect of romidepsin. No clinical response was observed in these seven patients. Authors speculated that the mechanism underlying resistance might be related to the compensatory induction of histone acetylation and chromatin changes. Furthermore, overall tolerability of the drug was considered limiting. The results of this study support the further evaluation of other HDAC inhibitors in combination with active therapies.

#### 4.2.2. HDAC Inhibitors with Molecular Targeted Therapies

Panobinostat was evaluated in combination with erlotinib in a phase Ib study for patients with HNSCC and NSCLC who had failed at least one line of systemic therapy [20]. The primary objective was to determine the maximum tolerated dose (MTD) of the combination treatment, with an expansion cohort of 20 patients at the recommended phase II dose. Paired pre- and post-treatment blood and fat pad biopsies to assess histone acetylation, as well as paired tumor biopsies to evaluate checkpoint kinase 1 (CHK1) expression as a pharmacodynamic biomarker, were obtained. Panobinostat was administered in combination with erlotinib in cycles of 21 days and for 2 out of the 3 weeks of each cycle. Panobinostat was given twice weekly orally, and erlotinib daily orally. Four dose levels were investigated in a 3 + 3 design: dose level 1 of panobinostat 20 mg and erlotininb 100 mg; dose level 2 of panobinostat 30 mg and erlotinib 100 mg; dose level 3 of panobinostat 30 mg and erlotininb 150 mg; and dose level 4 of panobinostat 40 mg and erlotinib 150 mg daily. Thirty-three patients were evaluable. The MTD of the combination therapy was determined at panobinostat 30 mg and erlotininb 100 mg (dose level 2), which was overall well tolerated. Adverse events included fatigue, nausea (grades 1–3), rash and anorexia (grades 1–2). Although the sample number was small (n = 7) for HNSCC patients, three out of seven patients achieved stable disease (SD), with a DCR of 43%, median OS of 8.2 months and median PFS of 2.1 months. [20]. Of the seven patients enrolled with squamous NSCLC, one had SD (14%), the median OS was 5.5 months and median PFS was 1.9 months. It is of interest to note that the DCR in HNSCC patients was greater compared to the DCR (14%) in squamous NSCLC. CHK1, a G2 m cell cycle regulator, has been shown to play a vital role in HDAC-inhibitor-mediated cytotoxicity in NSCLC cells and CHK1 overexpression is associated with resistance to HDAC inhibition. Interestingly, lower pre-treatment tumor protein expression levels of CHK1 (verified by immunohistochemistry) were associated with response to the panobinostat and erlotinib combination in six patients with NSCLC. The authors also evaluated the pharmacodynamic effect of panobinostat on global histone H4 acetylation levels (by Western blotting) in matching pre- and post-treatment abdominal fat pad biopsies and PBMCs from 17 patients. Increased protein levels of H4 acetylation were observed in 8 PBMC samples and 10 fat pad biopsies, with 7 of them overlapping. Of these patients, there were 4 matching tumor samples which also showed increased acetyl-tubulin levels (by IHC). Importantly, 67% of patients (8 out of 12) with a clinical response (SD or partial response) also had increased H4 acetylation levels in the fat pad biopsies, but only 36% of these patients showed increased H4 acetylation in PBMCs. Conclusively, this study suggests that the combination of panobinostat and erlotinib is well tolerated, and that CHK1 warrants further investigation as a predictive response biomarker. Furthermore, fat pad biopsies for H4 acetylation levels may be a rational approach to assess the pharmacodynamic effects of panobinostat. Although the sample size of HNSCC patients was low, the DCR of 43% in a previously pre-treated population draws attention towards the further investigation of panobinostat with or without erlotininb in HNSCC.

#### 4.2.3. HDAC Inhibitors with Chemotherapy

A few preclinical studies support the antitumor effect of HDAC inhibitors in combination with chemotherapy, which has led to ongoing clinical trials. A recent study [33] showed that valproic acid enhanced cisplatin-induced DNA damage through the downregulation of Excision Repair Cross-Complementing 1 (ERCC1) Excision Repair 1, which is critical in DNA repair, and by increasing cisplatin influx and decreasing cisplatin export from human HNSCC cancer cells. Treatment of HNSCC cells with valproic acid also decreased cisplatin- and/or cetuximab-induced nuclear translocation of EGFR, a mechanism known to render chemotherapy resistance. The synergistic antitumor effect of valproic acid in combination with cisplatin and cetuximab was confirmed in heterotopic and orthotopic HNSCC xenografts in nude mice [33]. Based on these findings, valproic acid is being evaluated in a phase 2 clinical trial (V-CHANCE) using valproic acid in combination with chemotherapy cisplatin and cetuximab in R/M HNSCC patients in the first-line setting [24] [NCT02624128].

#### 4.2.4. HDAC Inhibitors with Chemoradiotherapy

As described above, preclinical findings showed that vorinostat reverses cisplatin resistance in HPV-negative HNSCC cell lines and xenografts [31]. In addition, given the hypothesis that HDAC inhibition likely induces chromatin relaxation where platinum-based chemotherapy or radiation can induce DNA-damage more potently, vorinostat was evaluated in a phase 1 trial in combination with concurrent chemoradiation therapy in the treatment of advanced staged HNSCC [21]. Eligible patients had pathologically confirmed stage III, IVa, IVb disease that was unresectable or borderline resectable involving the larynx, hypopharynx, nasopharynx, and oropharynx. Vorinostat was started 1 week prior to the initiation of standard cisplatin and radiation therapy, and was continued throughout the chemoradiotherapy course. The primary objective of this study was to determine the MTD and safety of vorinostat in combination with standard chemoradiation therapy treatments in HNSCC. Vorinostat was given in a standard 3 + 3 dose escalation design, with doses ranging from 100 to 400 mg, three times weekly. Twenty-six patients met the eligibility criteria and completed the trial, 17 with HPV-positive and 9 with HPV-negative HNSCC. The MTD of vorinostat was determined at 300 mg every other day. The median follow up of enrolled patients was 33.8 months. The safety profile was promising, with anemia and leukopenia the most frequently identified adverse events, although all patients completed the chemoradiotherapy course without interruptions. A high rate of complete responses was reported (96.2%), with an estimated 5-year OS of 68.45% and 5-year disease-free survival of 76.6%, comparing favorably to historical controls of 70.9% and 46.2%, respectively. Although the study population was predominantly HPV-positive (17/26, 65.4%), the majority of the population had advanced disease (84.6% N2, N3) and significant smoking history (69.2% smokers), which are factors known to adversely affect the outcome of HPV-positive patients. Interestingly, vorinostat could also be administered through the G-tube, which is frequently required in patients with HNSCC receiving curative-intent chemoradiotherapy. Overall, this study reported high response rates with a toxicity profile comparable to the standard treatment of chemoradiotherapy, such as mucositis, xerostomia and dermatitis. Based on this study, a larger study investigating vorinostat in combination with standard-of-care chemoradiotherapy is planned.

In a similar concept, valproic acid was investigated in a phase 2 study in combination with standard platinum-based chemoradiotherapy in locally advanced HNSCC (NCT01695122) [23]. The primary objective of this study was to evaluate whether the addition of valproic acid increased the objective response rate in newly diagnosed patients with unresectable oropharyngeal or oral cavity HNSCC. Secondary endpoints included the safety and toxicity profile, PFS, OS and response rate based on HPV-status. Valproic acid treatment was initiated 2 weeks prior to the initiation of chemoradiotherapy at 15 mg/kg/day orally, it was up-titrated to a therapeutic plasma level of 40–100 ug/mL, and was continued until the completion of curative-intent treatment with cisplatin/RT. Due to significant toxicities, the trial was discontinued after 10 patients with HPV-negative oropharyngeal cancer were enrolled. Specifically, of the 10 patients treated, 3 patients were hospitalized with renal failure, respiratory infection and syncope, and 2 more patients experienced grade 3 and 4 adverse events with disseminated herpes zoster and radiodermatitis, respectively. The response rate at 8 weeks post-treatment in 9 evaluable patients was 88%. Biomarker assessment included PCR-based analysis of microRNAs (miRs) in the plasma and saliva of treated patients at baseline, 2 weeks after valproic acid treatment and 8 weeks after the completion of treatment. The investigators concluded that although the combination of valproic acid with cisplatin-based radiotherapy was associated with a high response rate, the toxicity rendered was prohibitory; thus, no further investigation of this combination seems prudent. Interestingly, a distinct pattern of miR expression was detected in responders versus non-responders, emphasizing the possible importance of specific miRs as diagnostic biomarkers of response to HDAC inhibition.

CUDC-101, which targets HDACs (class I, II), EGFR and HER2, has been evaluated in combination with chemoradiation in patients with HNSCC in a phase 1 study (NCT01384799) [25]. The primary objective was to determine the MTD of CUDC-101 in combination with cisplatin-based radiotherapy for patients with HNSCC. CUDC-101 was administered intravenously three times weekly for 1 week prior to the initiation of chemoradiotherapy, and then continued with the standard regimen of cisplatin-based radiotherapy. A total of 12 patients with intermediate- or high-risk HNSCC were enrolled, of which 11 patients were HPV-negative. The MTD of CUDC-101 was determined at 275 mg/m2/dose. In 5 of the 12 patients, CUDC-101 had to be discontinued due to adverse events; however, of these, only one was considered a dose-limiting toxicity. HDAC inhibition was observed in both PBMCs, tumor and skin biopsies. At 1.5 years of median follow up, one patient had recurrent disease, two patients died of causes not attributed to CUDC-101, and nine patients were free of progression. The investigators concluded that although the MTD was identified, there was a high rate of the DLT-independent discontinuation of CUDC-101, indicating the need to identify alternate schedules or routes of administration.

#### 4.2.5. HDAC Inhibitors with Immunotherapy

In addition to the role of HDAC inhibitors in sensitizing tumor cells to chemoradiotherapy, studies have also investigated the role of HDAC inhibitors in regulating immune-related genes, such as CD40 expression and HLA class I and II in different cancer cell lines [34,35,36]. Vorinostat or panobinostat, combined with immunomodulatory antibodies targeting CD40 and CD147 in mouse models of breast or colon adenocarcinoma solid tumors, induced complete tumor regressions with sustained immunological memory [37]. Treatment with HDAC inhibition induced tumor cell apoptosis, which induced the uptake of dead tumor cells by antigen-presenting cells (APCs), which then, in turn, activated CD8+ T-cell-mediated antitumor cytotoxicity. Another study showed that the treatment of breast and prostate carcinoma cells with clinically relevant doses of vorinostat or entinostat in vitro induced upregulation of the expression of a number of tumor-associated antigens, such as MUC1, brachyury and CEA, as well as antigen-processing machinery molecules. This reversed the immune evasion phenotype and enhanced the CD8+ T-cell-mediated lysis of cancer cells [38].

Extrapolating from preclinical studies such as those presented above, a phase II clinical trial with a safety lead-in cohort of pembrolizumab and vorinostat was pursued and recently completed (NCT02538510) in patients with R/M HNSCC and salivary gland cancer (SGC) [22]. Eligibility criteria included patients that had received any number of lines of therapy in the curative-intent or R/M setting, but no prior immunotherapy. A total of 25 patients with HNSCC and 25 with SGC were enrolled. Given that this review is focused on HNSCC, only the results pertaining to the HNSCC are discussed here. Of the 25 patients with HNSCC, 52% had p16+ oropharyngeal cancer. Pembrolizumab was given intravenously at 200 mg every 21 days, and vorinostat at 400 mg orally 5 days on and 2 days off during each 21-day cycle. This intermittent schedule was recommended by the sponsor of the study based on data suggesting better tolerability. Primary endpoints were safety and ORR. Secondary endpoints included OS and PFS. A proportion of 36% of R/M HNSCC had ≥ grade 3 adverse events. This safety profile was less favorable compared to pembrolizumab alone in the same patient population (13% of ≥grade 3 adverse events in Keynote-40). In the HNSCC cohort, 32% of patients had a PR and 20% had SD. These results are encouraging when compared to a historical control of approximately 20% PR with single-agent monoclonal anti-PD-1 antibodies in this patient population. The median overall survival (mOS) was 12.6 months and the median progression-free survival (mPFS) was 4.5 months in HNSCC. The mOS was 14.0 months and mPFS was 6.9 months. Overall, this study presented encouraging response rates in the HNSCC cohort with the combination of vorinostat and pembrolizumab, albeit with a less favorable toxicity profile compared to pembrolizumab alone. These results should be interpreted with caution, given that the HNSCC cohort was heterogeneous; it included cutaneous carcinomas which may have higher response rates to anti-PD-(L)-1 immunotherapy, and enrichment for higher PD-L1 expression could not be excluded. A larger study with a more homogeneous HNSCC population preselected for PD-L1 expression would be warranted to further investigate the efficacy of this promising combination regimen.

Another HDAC inhibitor, abexinostat, is being evaluated in combination with pembrolizumab in an actively recruiting phase 1b dose escalation study in patients with advanced solid tumors, including metastatic HNSCC (NCT03590054).

### 4.3. Histone Methylation/Demethylation in HNSCC

#### 4.3.1. Preclinical Data with Histone Methyltransferase Inhibitors in HNSCC

The methylation and demethylation of histones affect conformational changes of the nucleosome which are catalyzed by histone methyltransferases (HMTs) and histone demethylases (HDMTs). There are different lysine sites for methylation, such as K4, K9, K27, K36 or K79 of histone H3. The methylation of different lysine sites may induce transcriptional activation (H3K4me3, H3K79me3 or H3K36me3) or repression (H3K9me2, H3K9me3 or H3K27me3) [39]. A retrospective clinicopathologic analysis of HNSCC showed an association of high levels of H3K27me3 with advanced T status, N status, tumor stage, and perineural invasion, also associated with cancer-specific survival and disease-free survival [39].

EZH2, the catalytic component of the polycomb repressive complex 2(PRC2), is responsible for H3K27me3 and has been shown to play an important role in the development of HNSCC. High EZH2 protein expression has been observed in oral cavity HNSCC tumors and its expression has been shown to be correlated with poor survival [40,41]. Preclinical studies showed that EZH2 is also involved in regulating tumor growth, invasion and metastasis through H3K27me3 [42,43]. Another study showed that targeting EZH2 inhibits epithelial–mesenchymal transition (EMT) in HPV-negative HNSCC through downregulation of the expression of EMT-related markers such as N-cadherin and vimentin, but via upregulating E-cadherin [44].

In an in vitro study of HPV-negative HNSCC, EZH2 silencing was shown to potentiate cisplatin-based chemotherapy response [45,46]. The authors postulated a possible mechanism that EZH2 suppression results in a loss of chromatin condensation, which makes DNA more accessible to cisplatin and leads to more efficient DNA damage and cancer cell death.

In addition to the potential role of EZH2 inhibition in regulating tumor growth and metastasis, a recent preclinical study showed that targeting EZH2 may overcome anti-PD-1 resistance in HNSCC [47]. The authors of this study hypothesized that EZH2 inhibition may improve outcomes of anti-PD-1 therapy by enhancing antigen presentation in HPV-negative HNSCC. Analysis of 522 HNSCC HPV-negative tumors from TCGA showed a negative correlation between the EZH2 expression levels and HLA class I antigen-presenting molecules, including β2M, HLA-A, HLA-B, HLA-C and HLA-E. EZH2 inhibition resulted in a significant upregulation of HLA Class I expression in human and mouse HPV-negative HNSCC lines in vitro and in mouse models in vivo. EZH2 inhibitors or CRISPR-mediated EZH2 depletion increased antigen presentation in the tumor cells, and increased antigen-specific CD8+ T-cell proliferation, IFNγ production and tumor cell cytotoxicity. The authors showed that EZH2 inhibition increased antigen presentation through the reduction in histone H2K27me3 modification on the beta-2-microglobuin (β2M) promoter. In addition, combinatorial therapy of EZH2 inhibition and anti-PD-1 significantly suppressed tumor growth in an anti-PD-1 resistant model of HNSCC. This study provided preclinical evidence to further investigate EZH2 inhibition in combination with anti-PD-1 immunotherapy in HNSCC patients.

#### 4.3.2. Clinical Trials with EZH2 Inhibitors in HNSCC

Different EZH2 inhibitors, such as tazemetostat and CPI-1205, are currently being evaluated in clinical trials in multiple cancer types. In 2020, a phase 1/2 study was initiated, evaluating tazemetostat in combination with pembrolizumab in patients with R/M HNSCC. Eligibility criteria include: (1) R/M HNSCC, inclusive of cancers that originate in the head and neck region for the phase 1 part of the study; (2) R/M, PD-L1-positive HNSCC of the oral cavity, oropharynx, larynx or hypopharynx with the progression of disease on prior pembrolizumab or nivolumab treatment (monotherapy or chemoimmunotherapy) in the last 6 months for the phase 2 part of the study. The primary objectives are to determine the phase 2 recommended dose for the combination of tazemetostat with a fixed dose of pembrolizumab for the phase I part of the study, and the ORR for the phase 2 part of the study. Secondary endpoints include the incidence of adverse events, duration of response, PFS and OS. Tazemostat will be given orally twice daily on days 1–35 of cycle 1 (5-week cycle), then days 1–21 of subsequent cycles (3-week cycles). Pembrolizumab (200 mg) will be given intravenously at day 15 of cycle 1, then day 1 of each subsequent cycle. This study has recently started; therefore, no results have been reported yet. Efforts to investigate EZH2 inhibition in combination with anti-PD-1 immunotherapy in the first line setting for R/M HNSCC patients are also ongoing, but no studies have been initiated yet.

## 5. Conclusions

In this review, we have summarized the results from previous and ongoing clinical trials investigating epigenetic drugs in HNSCC. Although different epigenetic drugs have been investigated in multiple preclinical studies, either as monotherapy or combined with other anticancer agents, and have shown promising anti-tumor effects in HNSCC, very few phase 2 or disease-specific studies have been carried out to completion and/or have available results for review.

Regarding DNMT inhibitors, two phase 1 disease-specific studies investigating the combination of azacytidine with cisplatin were terminated due to accrual issues, thus still leaving the question open as to whether azacytidine may potentiate the effect of platinum-based chemotherapy in R/M HNSCC. An interesting study is ongoing to evaluate azacytidine as neoadjuvant monotherapy in HPV-positive HNSCC. Decitabine is currently being evaluated as monotherapy in R/M HPV-positive HNSCC in a phase Ib study, as well as in combination with durvalumab in the R/M HNSCC regardless of HPV status. However, no studies have evaluated decitabine in combination with platinum-based chemotherapy in R/M HNSCC. Overall, there still remain unanswered questions pertaining as to whether DNMT inhibitors may potentiate chemoradiotherapy in the curative-intent setting in select patients with HPV-positive or HPV-negative HNSCC, and whether a neoadjuvant approach, particularly in combination with immunotherapeutic interventions, may increase remission rates in the curative intent setting. Clinical investigation towards the above directions may be reasonable to pursue, albeit challenging by the lack of predictive biomarkers of response to DNMT inhibitors. Results from the NCT03019003 study of decitabine combined with durvalumab in patients with checkpoint refractory HNSCC are eagerly awaited.

Regarding HDAC inhibitors, a trial investigating romidepsin (NCT00084682) as monotherapy in the R/M HNSCC did not show clinical efficacy and tolerability was limiting; however, the expected pharmacodynamic effects with increased H3 hyperacetylation in PBMCs were observed, suggesting that other HDAC inhibitors with a better tolerability profile could be investigated in combination regimens in this patient population. In the curative-intent setting, the combination of valproic acid with cisplatin/RT was toxic and led to early termination of the NCT01695122 study. Similarly, the combination of CUDC-101 with cisplatin/RT (NCT01384799), although highly efficacious, was limited by a high rate of toxicities reported as independent to CUDC-101, requiring further safety evaluation. In contrast, the study by Teknos et al. [21] combining vorinostat with cisplatin/RT as a curative-intent therapy reported promising results with good tolerability and encouraging clinical activity, and has provided the stepping stone for a larger phase 2 study that is actively being pursued for HPV-negative HNSCC.

In the R/M setting, one key study by Rodriguez et al. [22] has investigated vorinostat in combination with pembrolizumab in PD-L1-positive, PD-(L)1 checkpoint-naïve HNSCC patients, with promising results and response rates higher (32%) compared to the historical control (20%). These results support further clinical investigation in a larger phase 2 study with a more homogeneous HNSCC population. Additional promising directions that merit further clinical investigation pertain to evaluating HDAC inhibition in the neoadjuvant setting in combination with immunotherapy, evaluating whether HDAC inhibition can potentiate chemoradiotherapy responses in the curative-intent setting, and deciphering the role of HDAC inhibition in potentiating chemoimmunotherapy responses in R/M HNSCC.

HNSCC, and particularly HPV-negative tumors, demonstrate genetic alterations leading to gene expression changes in protein methyltransferases and demethylases, with a significant body of preclinical evidence supporting the importance of this class of enzymes in the pathogenesis of this disease. A clinical trial investigating tazemetostat, an EZH2 methyltransferase inhibitor, was recently initiated in patients with R/M, PD-L1-positive HNSCC that have progressed on PD(L)-1 checkpoint blockade. A similar concept in the first-line, PD(L)-1 checkpoint-naïve setting is actively being pursued.

A major hurdle in implementing epigenetic interventions in clinical trials is that there are no available biomarkers of response to specific epigenetic interventions. Currently, there are no biomarkers that have been developed to predict responses to DNA-demethylating agents or histone deacetylase inhibitors. Investigating the specific mechanism of action of these drugs and finding potential biomarkers of clinical response in HNSCC is critical to select patients and formulate rational and successful clinical trial designs. Clinical trials in the neoadjuvant setting may enable the acquisition of valuable tumor specimens for the interrogation of mechanisms of action of epigenetic drugs.

Epigenetic interventions hold significant promise in the treatment of cancer, including HNSCC. The recent approval of tazemetostat by the FDA in the treatment of relapsed/refractory follicular lymphoma and locally advanced or metastatic epithelioid sarcoma [48,49] constitutes the latest achievement and highlights the promise of epigenetic therapies in cancer. To uncover specific mechanisms and effects mediated by epigenetic interventions in various cancer types, the application of novel epigenetic approaches in preclinical studies, such as the CRISPR-Cas9-mediated epigenetic editing, will be vital and expected to propel the field further ahead [50]. More potent and specific epigenetic drugs with favorable toxicity profiles are being developed, and preclinical work delineates mechanisms of action in HNSCC, although we envision the rational design of clinical trials that will target a select group of HNSCC patients with epigenetic vulnerabilities that can be targeted in combination with immunotherapy, chemotherapy and/or radiotherapy, rendering stronger and more durable responses while minimizing chronic and debilitating complications for patients with HNSCC.

## Figures and Tables

**Table 1 cancers-13-05241-t001:** Overview of clinical trials using DNMT inhibitors (monotherapy or combination) in HNSCC.

Reference/NCT	Status	Phase	DNMT Inhibitor	Chemotherapy/Immunotherapy	Study Duration	Disease Target	Result
NCT02178072	Recruiting	Window study	Azacytidine		2014-ongoing	HPV-positive HNSCC (resectable disease)	**Pending**
NCT04252248	Recruiting	1b	Decitabine		2019-ongoing	HPV-positive Anogenital and HNSCC (R/M)	**Pending**
NCT03019003	Recruiting	1b	Decitabine	Durvalumab	2017-ongoing	HNSCC (R/M, refractory to immune checkpoint blockade	**Pending**

Black, bold font: clinical trial ongoing and results not available yet.

**Table 2 cancers-13-05241-t002:** Drug Approvals and Examples of HDAC inhibitors.

Classification	Examples	* Specificity to HDAC	FDA-Approved Indication
Aliphatic fatty acids	Butyrate	Classes I and II	
	Phenylbutyrate	Classes I and II	
	Valproic acid	Classes I and II	
Hydroxamates	Vorinostat	Pan inhibitor	Cutaneous T-cell lymphoma
	Belinostat	Pan inhibitor	Peripheral T-cell lymphoma
	Givinostat	Pan inhibitor	
	Tefinostat	Pan inhibitor	
	Panobinostat	Classes I and II	Multiple myeloma
	Abexinostat	Classes I and II	
	Ricolinostat	Classes II	
	Pracinostat	Classes I, II and IV	
Benzamides	Entinostat	Class I	
	Mocetinostat	Class I	
	Tacedinaline	Class I	
	Domatinostat	Class I	
Cyclic peptides	Romidepsin	Class I	Cutaneous T-cell lymphoma
			Peripheral T-cell lymphoma
Sirtuin ligands	Nicotinamide	Class III	

* Specificity to HDAC: Class I (HDAC 1, 2, 3, 8), Class II includes lla (HDAC 4, 5, 7 and 9) and llb (HDAC 6, 10), Class lll (Sirtuins 1–7), Class IV (HDAC 11).

**Table 3 cancers-13-05241-t003:** Overview of clinical trials using HDAC inhibitors (monotherapy or combination) in HNSCC.

Reference/NCT	Status	Phase	HDAC Inhibitor	Chemotherapy +/− RT	Immunotherapy	Study Duration	Disease Site	Result
Haigentz et al. [19]/NCT00084682	Completed	2	Romidepsin			2004–2012	HNSCC (R/M)	-Confirmed pharmacodynamic effect of romidepsin -No objective responses
Gray et al. [20]	Completed	1	Panobinostat	Erlotinib		2008–2015	HNSCC and NSCLC (R/M)	-Combination was well tolerated -Small number of HNSCC patients -3/7 HNSCC patients achieved * SD, 43% * DCR -Higher number of HNSCC patients needed to evaluate adequate efficacy prior to considering a phase 2 study
Teknos et al. [21]	Completed	1	Vorinostat	Cisplatin/RT		2010–2019	HNSCC (locally advanced)	-Combination was well tolerated -Among 17 HPV+ and 9 HPV − HNSCC * CR (96%), estimated 5 yr * OS (68.45%) -Efficacy result warrants phase 2
NCT01267240	Terminated	2	Vorinostat	Capecitabine		2010–2017	HNSCC or * NPC (R/M)	No clinical activity
Rodriguez et al. [22]/NCT02538510	Active, not recruiting	2	Vorinostat		Pembrolizumab	2015–2020	HNSCC and Salivary gland cancer (R/M)	-Among 25 HNSCC patients, PR 32% -Toxicities higher than with pembrolizumab alone
Mak et al. [23]/NCT01695122	Completed	2	Valproic Acid	Cisplatin/RT		2012–2016	HNSCC (locally advanced)	Early termination due to toxicities
Caponigro et al. [24]/NCT02624128	Unknown	2	Valproic Acid	Cisplatin & Cetuximab		2015–unknown	HNSCC (R/M)	Not available
NCT03590054	Recruiting	1b	Abexinostat		Pembrolizumab	2018–ongoing	Advanced solid tumors	**Pending**
Galloway et al. [25]/NCT01384799	Completed	1	CUDC-101 (HDAC, EGFR, HER2 inhibitor)	Cisplatin/RT		2011–2018	HNSCC (intermediate/high risk)	-9/12 patients achieved CR -High rate of * DLT-independent discontinuation of drug warranting further phase I evaluation

* SD stable disease, DCR: disease control rate, OS: overall survival, SCC: squamous cell carcinoma, CR: complete response, PD: progressive disease, DLT: dose limiting toxicity, NPC: nasopharyngeal carcinoma. Blue, bold font: clinical trial completed and results available and interpretable. Black, bold font: clinical trial ongoing and results not yet available. Black font, not bolded: clinical trial terminated early or result not available.

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
