# Peer review of "Epigenetic Modifiers as Novel Therapeutic Targets and a Systematic Review of Clinical Studies Investigating Epigenetic Inhibitors in Head and Neck Cancer"

_cancers, 2021, doi:10.3390/cancers13205241_

Round 1
Reviewer 1 Report
The issue raised by the reviewer is resolved.
Author Response
No additional comments.
Reviewer 2 Report
The authors were thorough in their revision of this manuscript, and it is greatly improved. I appreciated the specific replies to my comments, and for a couple, the error was mine in missing the information or not connecting the dots as well as I should have. I apologize to, and thank, the authors for providing some clarification to me regarding those points, and why changes were not made. Not that it is important at this stage, but my comment about including "HNSCC" as a search term in the methods wasn't to replace "head and neck cancer" but in addition to. I agree with the authors that "head and neck cancer" would catch more studies, but since "head and neck squamous cell carcinoma" didn't seem to be used in addition to "head and neck cancer", I wasn't sure if any studies were missed as I have not done these searches. I do feel confident that the authors found the relevant studies needed for this review.
The additional sentences (and paragraphs) throughout the manuscript greatly improves the readability. The last paragraph, suggested by another reviewer, is a wonderful addition and way to conclude this review.
Well done!
Author Response
We appreciate the reviewer's evaluation!
Reviewer 3 Report
The article has been improved, but still some mistakes exist (mostly editorial):
- Unnecessary dot before the word ‘Introduction’
- '5-year overall survival (OS) remains low' --> please, be more precise
- 'standardof care' --> space missing
- ‘Overexpression of these DNMTs in different types of tumors results in hypermethylation’ --> hypermethylation of what? Please specify.
- O6-methyl-guanine-DNA methyltransferase --> 6 should be in superscript
- '20mg/m2' --> space missing
- 'objective response rate (ORR, , disease control rate (DCR, , quality of life, OS, assessed ≥ 6 months)' --> double commas? Please verify this sentence
- 'Another study using a pan-HDAC inhibitor, sodium phenylbutyrate, showed that it sensitizes response of HPV-negative HNSCC cells to cisplatin. Another study showed that sodium phenylbutyrate, a pan-HDAC inhibitor, sensitizes response of HPV-negative HNSCC cells to cisplatin' --> the two sentences are almost identical and repeat the information. Please rearrange.
- 'Table 3 summarizes the results of these clinical trials..'--> two dots
- 'These study results' --> check grammar
- 'panobinostat 20mg and erlotininb 100mg, dose level 2 of panobinostat 30mg and erlotinib 100mg, dose level 3 of pano-binostat 30mg and erlotininb 150mg, and dose level 4 of panobinostat 40mg and erlotinib 150mg daily'. --> spaces missing in drug doses!
- months. [27]. --> unnecessary dot
- ‘achieved SD’ --> the abbreviation defined later in the text --> ‘one had stable disease (SD)’ Please correct
- 'Based on preclinical work supporting that SAHA reverses cisplatin resistance in HPV-negative HNSCC cell lines and xenografts as described above [24], and also the hypothesis that HDAC inhibition likely induces chromatin relaxation where platinum-based chemotherapy or radiation can induce DNA-damage more potently, vorinostat was evaluated in a phase 1 trial in combination with concurrent chemoradiation therapy in the treatment of advanced staged HNSCC' -> the sentence is too long; also unify if you use the drug name ’SAHA’ or ‘vorinostat’
- '100-400mg' --> again space missing!
- '15mg/kg/day'
- 'radiodermatitis respectively' --> comma missing
- '275mg/m2/dose'
- '200mg'
- 'HNSCC The' --> dot missing
- 'complex 2(PRC2)' --> space missing
- 'CRISPR mediated' --> CRISPR-mediated
- '200mg'
- 'DNA methyltransferase' --> DNMT
Author Response
Please see the attachment.

This manuscript is a resubmission of an earlier submission. The following is a list of the peer review reports and author responses from that submission.
Round 1
Reviewer 1 Report
The reviewed manuscript entitled ‘Epigenetic modifiers as novel therapeutics and a systematic review of clinical studies investigating epigenetic inhibitors in head and neck cancer’ aims to provide a comprehensive summary of the current knowledge on the function of different epigenetic modifiers in HNSCC. In this review the clinical implications of epigenetic modifications as well as the results of clinical trials evaluating epigenetic interventions are discussed.
The review is subdivided into sections devoted to DNMT inhibitors, HDAC inhibitors and bromodomain inhibitors and their respective clinical trials. Each part is divided into pre-clinical and clinical data. Both parts are described with the relevant details (clinical trials) and appropriate number of examples (pre-clinical studies), however, the introduction to each epigenetic mechanism covered is very short and kind of trivial. Taking into consideration that the article should not be too long I suggest improving each introduction by adding few more sentences in each part and also citing relevant papers guiding the interested readers for further reading.
The paper contains interesting data and generally deserves publication (not sure if the high standards of Cancers are being met), but some improvements need to be implemented first. Both Introduction and Conclusions need to be improved. In the Introduction some repetitions occur, while Conclusions mostly summarize the previously cited studies/trials. It would be valuable to mention the latest achievements in epigenetic therapies, and the novel approaches such as CRISPR-Cas9 mediated epigenetic editing. However, taking into consideration that no clinical trials are being performed based on these approaches yet I think only mentioning it in the Conclusions and referring to another review article [PMID: 33572577] would be satisfactory. Generally, more mature commentary in both above mentioned sections is required.
The article section titles should be corrected. For instance section 4. is Histone modifications, and 5. Is Histone methylation/demethylation (being also histone modification, so should be introduced as a subsection). Moreover section ‘DNA methylation/demethylation’ is also confusing, as only demethylating drugs are being used.
Moreover, several other mistakes must be corrected:
- The authors use either the term: ‘head and neck squamous cell carcinoma’ or ‘head and neck cancer’ or the abbreviation: ‘HNSCC’. The term ‘head and neck squamous cell carcinoma’ should be introduced at the beginning and then only the abbreviation should be used throughout the whole text.
- For gene names italics should be used. Please correct.
- ‘…oropharyngeal squamous cell carcinoma patients (OSCC, HPV status was not specified)’ --> should be ‘…oropharyngeal squamous cell carcinoma patients (OSCC). HPV status was not specified in this study.’
- ‘DAP-K’ --> should be ‘DAPK’
- 'Chen et al’ --> dot missing ‘et al.’ Check the entire manuscript and correct wherever necessary.
- ‘5-aza-2’-deoxycytidine’, ‘DAC’ or ‘decitabine’ are used interchangeably. Same refers to ‘5-azacytidine’ and ‘ azacytidine’. Please unify as it can be confusing for the readers. Check the entire text for similar mistakes.
- 20mg/m2 --> 20 mg/m2
- ‘ORR (at 3 and 6 months), DCR (at 3 and 6 months), quality of life (at week 3, 5, 8 and 24) and OS’ --> the abbreviations should be defined when they are first used
- “plasticity’ --> the quotation marks should be unified
- 'The authors concluded that HDAC inhibition suppress the invasive and migratory potential of HPV-negative through disruption of the EGFR-Arf1 complex pathway --> should be ‘…of HNSCC…’
- ‘determine the MTD’ --> define what MTD stands for
- ‘panobinostat 20mg and erlotininb 100mg’ --> introduce spaces: 20 mg, 100 mg etc.
- ‘33 patients’ --> when starting a sentence with the number: ‘Thirty-three patients’
- ‘3 out 7 patients’ --> ‘3 out of 7 patients’
- (IHC) --> e.g. ‘(verified by immunihistochemisty)’
- ‘pat pad biopsies’ --> fat pad biopsies’
- ‘ERCC1 Excision Repair 1’ --> ‘Excision repair cross-complementing 1 (ERCC1)’
- For clarity, define the abbreviation: ‘R/M’
- ‘daily daily’
- ‘toxicity profile of capecitabine with vorinostatPFS’ --> correct
- 25 patients
- vorinostat (SAHA) --> leave only one, previously introduced name
- ‘70.9% and 46.2% respectively’ --> comma missing
- ‘through the G-tube through a novel formulation’ --> please clarify
- ‘HNSCCC’ --> remove the additional C
- ‘15m/kg/day’ --> correct
- ‘40-100ug/ml --> ‘40-100 µg/ml’
- ‘polymerase chain reaction of’ --> ‘PCR-based analysis of’
- At 1.5 years of median follow up, one patient had recurrent disease, two patients died of causes not attributed to CUDC-101, and 9 patients were free of progression. --> please unify how numbers are written.
- in Keynote-40)In the --> space and dot missing. Check the entire text for other similar mistakes.
- ‘epithelial mesenchymal transition’ --> ‘EMT’
- Conclusions – replace the repeated word: ‘setting’ with another one
- ‘plethora of genetic and expression alterations’ --> rewritte e.g. ‘genetic alterations leading to gene expression changes’
- ‘EZH2 methyltransferase enzymatic inhibitor’ --> the word ‘enzymatic’ should be removed
- Check the manuscript for any double-space mistakes. There are plenty of such.
Reviewer 2 Report
This review, “Epigenetic modifiers as novel therapeutics and a systematic review of clinical studies investigating epigenetic inhibitors in head and neck cancer” provided a summary of preclinical experiments and clinical trials using epigenetic modifiers, either alone or in combination with more standard treatment regimens. Overall it is an interesting review and of interest to clinicians, but I feel that this review needs major revision before it can be published. My suggestions are below.
Major criticisms with this review:
- Substantial detail was provided about the parameters for many of the clinical trials, but often there were no results available after an extensive description of the clinical trial (for example, after paragraphs #1 and #2 in section 3.2.2 – is all this detail needed in a review of results??) or the authors summarize the conclusion of the clinical trial without providing any data (for example on top of page 4, “Preliminary results from the analysis of 5 HPV-positive tumors from patients participating in this window of opportunity study showed that after 5 or 7 days of treatment, azacytidine decreased the expression of HPV genes, stabilized and increased the expression of p53, and induced activation of caspase 3 and apoptosis in HPV-positive HNSCC tumors.” Is this decrease in expression of HPV genes significant? What was the fold change? The reader is left with a general conclusion but not with any concrete data. In some places, the authors did provide percentage of expression (e.g., 32% vs 20%) which is appreciated.
- There were many abbreviations that were used and not defined at the first mention. For some abbreviations, no definition was given. For others, the abbreviation was used earlier on, and then later, the abbreviation was spelled out, and then other places the abbreviation was not used and just the spelled out version was used. Many examples: ORR, DCR, OS, PFS, MTD, etc. Eventually the reader will find what these stand for (although I never found MTD = maximum tolerated dose), but not at the first use. I realize that many of these are common abbreviations in papers reporting clinical data and outcomes, but consistency would be appreciated. It would be useful to include a glossary or table of these terms, if this format would be accepted by the journal.
- The paper would be greatly improved if the sections were reorganized to discuss the clinical trials in terms of their results instead of by study. Additionally, it would be helpful to add a section to summarize the results of the clinical trials based on outcome. As it is, the reader is left having to synthesize which trials were successful, which weren’t, which were done on HPV-positive HNSCC, which were done on HPV-negative HNSCC, which included other cancers (such as salivary cancer), etc. It seemed as if each section was written like this: Preclinical data or a clinical trial, it’s parameters, and generalized results; there wasn’t any cohesiveness to tie them together.
Minor criticisms with the review:
- There are many typographical or grammatical errors throughout; too many/difficult for me to provide a list of all of them here without being able to refer to line numbers. Please proofread carefully.
- The sentence “DNMT1 overexpression was significantly associated with the overall survival and relapse-free survival of patients.” Implies that it is positively correlated, but the next sentence indicates that it is negatively correlated. This was done in several places – I’d suggest authors be more clear when using “correlated” or “associated” by providing phrasing such as “inversely correlated” or “significantly associated with an increased risk of relapse”.
- The authors did not include the search term “HNSCC” in their methods. Is this needed? Were studies missed, or would their search term “head and neck cancer” catch this? Would be worth checking to see if studies were missed.
- This sentence near the end of section 3.2.1 “In contrast, a tumor from one HPV-negative patient showed decreased expression of p53 expression, while MMP expression was increased after treatment with azacytidine [11]” was put in a paragraph that focused on HPV-positive HNSCC. This seems out of place, and if the authors want to include it, more discussion on this needs to be included. Again, there’s general results but no way for the reader to evaluate if these findings on one patient sample is significant or not.
- I would like the authors to expand on this sentence (paragraph 2, page 6): “This study suggests that HDAC inhibition may affect tumor “plasticity’ and thereby create a homogeneous population of cancer cells by selecting out a subpopulation of cancer cells involved in tumor progression and the development of resistance to therapy.” I find it hard to believe that reduced H3K9 acetylation using an HDAC inhibitor would create a homogeneous population of cancer cells. HDAC inhibitors can affect many genes, and as such, without an analysis of these tumor cells, this seems like a statement that can’t be made. However, if the authors can provide more context as to what was meant, that would be helpful.
- What is meant by (paragraph 3, page 6): “Another study using a pan-HDAC inhibitor, sodium phenylbutyrate, showed that it sensitizes response of HPV-negative HNSCC cells to cisplatin by disrupting the Fanconi anemia/BRCA pathway [19].” There are several of these types of sentences that, to me, are vague since the data, or a summary of the data, are not provided in this review. Another one is (top of page 9): Overall, this study reported high response rates with a toxicity profile comparable to the standard treatment of chemoradiotherapy.” What is the response rate? What is the toxicity profile compared to standard treatment?
- From this information (bottom of page 7): “Importantly, 67% of patients (8 out of 12) with a clinical response (stable disease or partial response) also had increased H4 acetylation levels in the pat [typo: fat] pad biopsies, but only 36% of these patients showed increased H4 acetylation in PBMCs. These data suggest that the combination of panobinostat and erlotinib is well tolerated, and that CHK1 warrants further investigation as a predictive biomarker of response.” The authors do not explain how these data show that this combination therapy is well-tolerated, or how CHK1 is a biomarker.
- HMTs are better referred to as histone methyltransferases, not histone methylases (section 5.1).
- I am unsure what this statement means in the context of this review (last paragraph, section 4.2.4): “HDAC inhibition was observed in both peripheral blood mononuclear cells (PBMCs), tumor and skin biopsies.” The authors mention PBMCs in more than one place but I never did understand the connection with HNSCC cells and PBMCs. In the conclusion, the authors suggest “…the expected pharmacodynamic effects with increased H3 hyperacetylation in PBMCs were observed, suggesting that other HDAC inhibitors with a better tolerability profile could be investigated in combination regimens in this [HNSCC] patient population.” What is the connection between acetylation in PBMCs vs epithelial cells?
Reviewer 3 Report
This review summarizes different epigenetic drugs in head and neck cancer and their clinical implications in ongoing clinical trials. I consider the topic relevant as it is important to know how epigenetic drugs are doing in clinical trials. It shows past/ongoing clinical trials for epigenetics drugs in head and neck cancer that has not been reviewed recently. The authors specify their methodology clearly to perform their review. The conclusions consistent with the evidence and arguments presented and they address the main question posed. But references are low and could have more references and present different articles. It is a very informative review and I recommend to be published in the present form.
Reviewer 4 Report
The manuscript by Burkitt et al. very nicely sums up the subject of epigenetic therapeutics in head and neck cancer in the aspect of preclinical and clinical studies. The tables are representative and informative. In the opinion of the reviewer the shown material is an accurate overview of the subject. Therefore, this reviewer support is publication in Cancers. Minor points: The font and resolution of all Tables should be improved. I recommend to separate the tables from the text.Author Response
Please see attachment.
